# A 1-Dimensional Sympagic–Pelagic–Benthic Transport Model (SPBM): Coupled Simulation of Ice, Water Column, and Sediment Biogeochemistry, Suitable for Arctic Applications

**Shamil Yakubov [1,\*], Philip Wallhead [2], Elizaveta Protsenko [3,4,\*], Evgeniy Yakushev [3,4,\*], Svetlana Pakhomova [4] and Holger Brix [1]**

[1]   Institute of Coastal Research, Helmholtz-Zentrum Geesthacht (HZG), Max-Planck-Straße 1, 21502 Geesthacht, Germany
[2]   Norwegian Institute for Water Research (NIVA vest), Thormøhlensgate 53 D, 5006 Bergen, Norway
[3]   Norwegian Institute for Water Research (NIVA), Gaustadalléen 21, 0349 Oslo, Norway
[4]   P.P.Shirshov Institute of Oceanology RAS, Nakhimovskiy prosp. 36, Moscow 117991, Russia
\*   Correspondence: shamil.yakubov@hzg.de (S.Y.); Elizaveta.Protsenko@niva.no (E.P.); evgeniy.yakushev@niva.no (E.Y.)

**Abstract:** Marine biogeochemical processes can strongly interact with processes occurring in adjacent ice and sediments. This is especially likely in areas with shallow water and frequent ice cover, both of which are common in the Arctic. Modeling tools are therefore required to simulate coupled biogeochemical systems in ice, water, and sediment domains. We developed a 1D sympagic–pelagic–benthic transport model (SPBM) which uses input from physical model simulations to describe hydrodynamics and ice growth and modules from the Framework for Aquatic Biogeochemical Models (FABM) to construct a user-defined biogeochemical model. SPBM coupled with a biogeochemical model simulates the processes of vertical diffusion, sinking/burial, and biogeochemical transformations within and between the three domains. The potential utility of SPBM is demonstrated herein with two test runs using modules from the European regional seas ecosystem model (ERSEM) and the bottom-redox model biogeochemistry (BROM-biogeochemistry). The first run simulates multiple phytoplankton functional groups inhabiting the ice and water domains, while the second simulates detailed redox biogeochemistry in the ice, water, and sediments. SPBM is a flexible tool for integrated simulation of ice, water, and sediment biogeochemistry, and as such may help in producing well-parameterized biogeochemical models for regions with strong sympagic–pelagic–benthic interactions.

**Keywords:** arctic; biogeochemical modeling; transport model; ice; sediments

## 1. Introduction

Arctic marine ecosystems have undergone drastic changes and the most important changes are climatically driven [1–5]. The Coupled Model Intercomparison Project and the community climate system model studies have projected atmospheric warming in the Arctic of 1.5–4.5 times the mean global warming, and the Arctic marine environment is expected to be strongly impacted by a loss of ice cover, increasing light exposure, ocean warming, freshening, acidification, and deoxygenation [6]. Modeling simulations are needed for the analysis of present conditions and the projection of long-term impacts on Arctic marine biogeochemistry.

A biogeochemical model suitable for the Arctic should take into account the specific conditions of this region, such as the seasonal to permanent ice cover and the presence of shelf areas. Thus, the model should preferably combine processes occurring in three domains: ice, water column, and sediments. Each of these domains has some specific features and modeling challenges:

*Ice.* The Arctic ice-algal primary production is a significant part of the total primary production of the Arctic region [7]. Photosynthetic microorganisms extend the production season, provide a winter and early spring food source, and contribute to organic carbon export to depth [8]. A modeling study [9] estimated an average Arctic ice-algal primary production of 21.7 Tg C year $^{-1}$, which equates to roughly 5% of total pelagic primary production [10] for this area. Other authors [7] estimated sea ice-algal production accounting for 5–10% of total Arctic and Southern Ocean primary productivity. Another modeling study [11] suggested that under a mild climate change scenario the sea ice community around Greenland may become generally more productive while pelagic phytoplankton productivity may decrease. It is therefore desirable to include the ice domain in biogeochemical modeling studies of the Arctic region. There are three main approaches to implement ice algae behavior according to the place where algae live in the ice column [12,13]: in the bottom layer of an ice column with fixed thickness, in the bottom layer of an ice column with variable thickness, or in any layer of an ice column. Recent research suggests that ice-algal models should resolve the ice vertically to avoid biases that may result from either assuming that ice algae are solely present at the bottom layer or that they have a homogeneous vertical distribution [10].

*Water column.* In the Arctic, global climate change is causing seawater acidification, accompanied by local changes in productivity and oxygen depletion [14,15]. It follows that the carbon cycle can be an important component of multidecadal-scale biogeochemical models. Oxygen dynamics and redox process parameterization can also be useful in areas affected by oxygen depletion (often in estuaries and fjords). To improve the representation of near-bottom processes the benthic boundary layer (BBL) should be resolved within the water column domain. The BBL is "the part of the marine environment that is directly influenced by the presence of the interface between the bed and its overlying water" [16]. For the Arctic, this layer is especially important since ice melting and permafrost thawing can drive strong fluxes of ungrazed organic material to the BBL [17].

*Sediments.* Sinking fluxes from the water column can provide sources of new energy for the benthic community. Also, it has been shown [18] that benthic, as well as pelagic, activity can be an important factor for annual pH variability in coastal areas. Sediment layers in models should therefore respond accurately to sinking fluxes and provide accurate remineralization rates. Redox processes occurring in sediments can be highly structured in the vertical direction [19], suggesting a need for explicit vertical resolution in sediment models.

In view of these features and challenges, we aimed to develop a flexible 1D vertical transport model that, when coupled with a biogeochemical model, can provide integrated simulation of biogeochemical processes in ice, water column, and sediment domains, with a vertically-resolved grid for each. The resulting sympagic–pelagic–benthic model (SPBM) uses NetCDF file inputs from hydrodynamic/ice models to describe an "offline" physical environment, and the Framework for Aquatic Biogeochemical Models (FABM) [20] to provide biogeochemical source-minus-sink terms and vertical sinking velocities. FABM is "a Fortran 2003 programming framework for biogeochemical models of marine and freshwater systems. FABM enables complex biogeochemical models to be developed as sets of stand-alone, process-specific modules." (FABM wiki). The FABM coupling allows the user to construct their own biogeochemical model using existing modules in the FABM library plus any new modules written by the user (SPBM does not itself provide any new biogeochemical modules). The FABM library is rapidly expanding and presently includes modules from some of the most detailed published biogeochemical models, e.g., The European regional seas ecosystem model (ERSEM) [21], the bottom-redox model biogeochemistry (BROM-biogeochemistry) [22], the PCLake aquatic ecosystem model [23], and the model for adaptive ecosystems in coastal seas (MAECS) [24,25]. As with FABM, SPBM transport code is written in FORTRAN.

The paper is structured as follows: Section 1—Introduction (this part); Section 2—Description of the SPBM routines; Section 3—Results from two test simulations to demonstrate SPBM's capabilities and its relevance to Arctic biogeochemical modeling; Section 4—A discussion of SPBM capabilities and limitations; Section 5—Conclusions.

## 2. Methods—A 1D Transport Model

SPBM is a 1D advection–diffusion–reaction solver that uses FABM to define an arbitrary biogeochemical model structure and to calculate reaction terms, sinking speeds within the water domain, and various optional biogeochemical diagnostics. FABM distinguishes three types of model variables: state variables, diagnostic variables, and dependencies. State variables are the basic elements for which the rates of changes must be provided (e.g., nitrate, chlorophyll concentrations). Diagnostic variables are calculated within FABM according to the values of the state variables and dependencies at each time step (e.g., pH, nitrification rate). Dependencies are the physical environment variables and interconnections within FABM (e.g., temperature, salinity). SPBM sends dependencies to FABM and updates the state variables over each time step using various advection/diffusion algorithms and the FABM-calculated reaction terms. SPBM outputs all necessary state and diagnostic variables in NetCDF files. Within SPBM, state variables are considered as solute or particulate concentrations.

### 2.1. Formulation and Numerical Integration

SPBM solves a system of 1-D transport equations in Cartesian coordinates for all three domains (ice, water column, and sediments). The dynamics are

$$\frac{\partial C_i}{\partial t} = \frac{\partial}{\partial z} A_f D \frac{\partial C_i P_f}{\partial z} - \frac{\partial}{\partial z} u C_i + R_i \tag{1}$$

where $C_i$ is the *i*-th state variable in units provided by the biogeochemical model through FABM, (mmol m$^{-3}$ total volume) or (mg m$^{-3}$ total volume) (here total volume refers to a representative control volume including both liquid and solid); t is the time step, (s); z is the depth, (m); $A_f$ is the porosity-related area restriction factor for fluxes, dimensionless; D is the total diffusivity, (m$^2$ s$^{-1}$); $P_f$ is the porosity factor, dimensionless; u is the sinking velocity (advection/burial in the sediments), (m s$^{-1}$). $R_i$ is the combined sources minus sinks of the *i*-th state variable provided by the biogeochemical model through FABM, (mmol m$^{-3}$ total volume s$^{-1}$) or (mg m$^{-3}$ total volume s$^{-1}$). The porosity factor $P_f$ is used to calculate the volume concentration in brine (in the ice column) or in pore water/solid matrix in the sediments. Exchange within the ice and sediment layers occurs through brine channels and through pores or solid matrix, so the area restriction factor $A_f$ is included to limit fluxes within the respective phases (intraphase mixing). The values of $A_f$, $P_f$, D, and u depend on whether these parameters are calculated in ice, water column, or sediment domains and whether the state variable is solute or particulate.

*In the ice domain:*

For particulates, it is assumed that the concentration is the same in both the brine channels and ice matrix, hence $P_f = 1$. However, vertical fluxes are assumed to be restricted to the brine channels where the particulates are mobilised in suspension, hence $A_f = \varphi(z)$. Here, the dimensionless porosity $\varphi(z)$ is equal to the relative volume of the brine channels in the ice [26], which can be obtained from an ice thermodynamic model or using empirical relationships (see Appendix A). Solutes are assumed to be excluded from the ice matrix, hence $P_f = \frac{1}{\varphi(z)}$, and fluxes are again restricted to the brine channels, hence $A_f = \varphi(z)$. The total diffusivity D in the ice brine channels is a sum of the molecular diffusivity $D_m(s)$ (m$^2$ s$^{-1}$) on the ice–water interface (applied only to solutes), the gravity drainage diffusivity

$D_{gd}(z)$ $(m^2 s^{-1})$ at depths z within the ice, and the diffusivity caused by convection that occurs in the bottom layer of the growing ice $D_{gi}(s)$ $(m^2 s^{-1})$ [26]:

$$D = D_m(s) + D_{gd}(z) + D_{gi}(s)$$
$$D_{gd}(z) = F_{vb}z_b$$
$$D_{gi}(s) = 10^{-2}z_s(9.667 \cdot 10^{-9} + 4.49 \cdot 10^{-6}IceGrowth - 1.39 \cdot 10^{-7}IceGrowth^2)$$

where s means that the value of the parameter is determined only on the interface between the bottom (skeletal) layer of ice and surface water layer; $F_{vb}$ is a constant mean flux volume rate from the brine channels, $(m s^{-1})$; $z_b$ is the vertical distance over which the ice column is influenced by brine tube convection (depths where $\varphi(z) > \varphi_{min}$), (m); IceGrowth is the total ice growth rate $(cm s^{-1})$; $z_s$ is the thickness of the ice layer, (m). $D_{gi}(s)$ is not equal to zero only during the period of ice build-up and only on the interface between water and ice. Alternatively, the total diffusivity D can be read from an input file generated by e.g., an ice thermodynamic model.

The sinking velocity u is non-zero only for particulate variables in the layers where $\varphi(z) > \varphi_{min}$ (if $\varphi(z) \leq \varphi_{min}$ sea ice brine pockets are not interconnected) and is generally determined at each time step by the biogeochemical model through FABM. For all diatoms living in the ice column, to represent their ability to maintain their vertical position relative to the skeletal layer [26], u is set to a constant but possibly layer-dependent value within the ice column and zero on the ice–water interface between ice and water domains, while the total diffusivity D is set to zero.

*In the water column domain:*

Here $P_f = 1$ and $A_f = 1$ at all depths for both solutes and particulates, since there is only one phase to consider.

The total diffusivity D is composed of the molecular diffusivity $D_0$ $(m^2 s^{-1})$ (applied only to solutes) and the turbulence diffusivity $D_t(z)$ $(m^2 s^{-1})$:

$$D = D_0 + D_t(z)$$

where $D_t(z)$ is taken from the hydrophysical model as input data. The water column domain contains the structure that could be called the BBL. It is located in the lower part of the water column. Turbulent diffusivity for each layer $z_i$ within the BBL is linearly decreasing from the deepest non-zero value of the diffusivity $D_t(z_d)$ as follows:

$$D_t(z_i) = \frac{D_t(z_d)}{z_d - z_0}(z_i - z_0)$$

where $z_d$ (m) is the deepest depth with non-zero value of $D_t(z)$ and $z_0$ (m) is the bottom depth.

The sinking velocity u is taken from the biogeochemical model through FABM for all particulates and is zero for all solutes.

*In the sediment domain:*

Within the sediments, particulate variables are confined to the sediment matrix and solutes are confined to the pore water. So, for solid particulates the porosity factor $P_f = \frac{1}{1-\varphi(z)}$ and the area restriction factor $A_f = 1 - \varphi(z)$ at depths z. For solutes $P_f = \frac{1}{\varphi(z)}$ and $A_f = \varphi(z)$. There is no adsorption in the present version.

A time-independent porosity $\varphi(z)$ at depths z through the entire sediment domain is described following [27]:

$$\varphi(z) = \varphi(z_\infty) + (\varphi(z_0) - \varphi(z_\infty))e^{-\frac{(z-z_{swi})}{k_\varphi}}$$

where $\varphi(z_\infty)$ is the deep porosity, dimensionless; $\varphi(z_0)$ is the porosity at the sediment–water interface (SWI), dimensionless; $z_{swi}$ is the depth of the SWI, (m); $k_\varphi$ is the coefficient for exponential porosity change, (m).

The total diffusivity D is a sum of the molecular diffusivity $D_m(z)$ $(m^2\,s^{-1})$ (applied only to solutes) and the bioturbation diffusivity $D_b(z)$ $(m^2\,s^{-1})$ [28]:

$$D = D_m(z) + D_b(z)$$
$$D_m(z) = D_0 \frac{1}{1-2\ln\varphi(z)}\mu_d$$
$$D_b(z) = D_{bo}(z)\frac{O_2}{O_2 + K_{O_2}}$$

where $D_0$ is the infinite-dilution molecular diffusivity, $(m^2\,s^{-1})$; $\mu_d$ is the relative dynamic viscosity, dimensionless; $O_2$ is the oxygen concentration in the bottom layer of the water column, (mmol m$^{-3}$); $K_{O_2}$ is the half-saturation constant, (mmol m$^{-3}$). The oxygen-saturated bioturbation diffusivity [22] $D_{bo}(z)$ $(m^2\,s^{-1})$ depends on the distance $z_{db}(z)$ (m) between the interface depth z and the depth with a constant bioturbation activity as follows:

$$z_{db}(z) = z - (z_{swi} + z_{cb})$$
$$\text{if } z < z_{swi}+z_{cb}: D_{bo}(z) = D_{bm}$$
$$\text{if } z > z_{swi}+z_{cb}: D_{bo}(z) = D_{bm}e^{-\frac{z_{db}(z)}{F_d}}$$

where $z_{swi}$ is the depth at the SWI, (m); $z_{cb}$ is the constant bioturbation activity layer thickness, (m); $D_{bm}$ is the maximum bioturbation diffusivity, $(m^2\,s^{-1})$; $F_d$ is the bioturbation decay scale, (m).

On the SWI it is assumed that the bioturbation diffusivity mixes concentrations in units (mmol m$^{-3}$ total volume) instead of (mmol m$^{-3}$ solids/solutes) (interphase mixing). Therefore, special values of $P_f$ are needed for the layers immediately above and below the SWI (see Appendix B):

$$\text{for solutes}: P_f(z_{a,b}) = \frac{\frac{\varphi_{swi}}{\varphi_{a,b}}D_m(z_{swi}) + D_b(z_{swi})}{\varphi_{swi}(D_m(z_{swi}) + D_b(z_{swi}))}$$
$$\text{for solids}: P_f(z_{a,b}) = \frac{1}{1 - \varphi_{swi}}$$

where the subscripts a, b and swi determine a location of the corresponding variables: a means the layer above, b the layer below, swi on the SWI.

The advection/burial velocities $u(z)$ are described following [22]:

$$\text{for solutes}: \; u(z) = \frac{\varphi(z_\infty)}{\varphi(z)}u_b + \frac{1}{\varphi(z)}D_b^{inter}\frac{\partial\varphi(z)}{\partial z} \tag{2}$$

$$\text{for particulates}: \; u(z) = \frac{1-\varphi(z_\infty)}{1-\varphi(z)}u_b - \frac{1}{1-\varphi(z)}D_b^{inter}\frac{\partial\varphi(z)}{\partial z} \tag{3}$$

where $u_b$ is the deep burial velocity, (m s$^{-1}$); $D_b^{inter}$ is the interphase component of the total bioturbation diffusivity $D_b = D_b^{intra} + D_b^{inter}$, and $D_b^{inter}$ is nonzero only on the SWI where $D_b^{inter} = D_b$ (m$^2$ s$^{-1}$). Note that although a non-zero $D_b^{inter}$ beyond the SWI would alter the computed advection/burial velocities $u(z)$ via Equations (2) and (3), the net transport of biogeochemical tracers would not be affected because corresponding interphase components $\frac{1}{\varphi}C_i D_b^{inter}\frac{\partial\varphi}{\partial z}$ and $\frac{1}{1-\varphi}C_i D_b^{inter}\frac{\partial(1-\varphi)}{\partial z}$ would need to be added to the diffusive fluxes in Equation (1), and these will exactly cancel the contributions of $D_b^{inter}$ to advection/burial. In other words, when the porosity profile is specified and used to compute advection/burial velocities under steady state compaction, the tracer advection/diffusion depends only

on the total bioturbation diffusivity, and the intraphase form assumed by SPBM for diffusion inside the sediments is correct irrespective of the relative contribution of inter- vs. intraphase mixing, see [29]. However, there can be no intraphase component of a Fickian particulate diffusion across the SWI because by definition there is no solid matrix above the SWI [22].

Equation 1 is integrated numerically over a single combined (ice, water column, sediments) grid, using a constant model time step. The coupling method follows an operator splitting approach [30]: concentrations are successively updated by contributions over one time step of diffusion, reaction, and sinking/advection/burial, in that order. Diffusive updates are calculated by a semi-implicit central-space algorithm adapted from a routine in BROM-transport [22] which in turn was adapted from the general ocean turbulence model (GOTM) [31]. Sinking/advection/burial updates are calculated using a first-order upwind differencing scheme. Reaction updates are calculated from forward Euler time steps.

### 2.2. The Grid

SPBM uses a fixed grid structure for the water column and sediments, and a time-dependent grid for the ice column. The number of grid points inside the ice column can vary with time but the spacing is fixed (see Figure 1). Water column layer depths (m) are taken as input from a hydrophysical model (distances between layers can be unequal) and extra layers are incorporated in the lower part in order to fully resolve the BBL. Total ice thickness (m) for every day of simulation is also taken as input from a hydrophysical model, and the ice column is constructed using a fixed layer thickness (m) as an input parameter. Therefore, the ice column is discretized into layers of strictly constant thickness $z_s$, and when the ice column grows or melts its total thickness can change only by multiples of $z_s$. This simplification facilitates recalculation of the variable concentrations during melting and freezing.

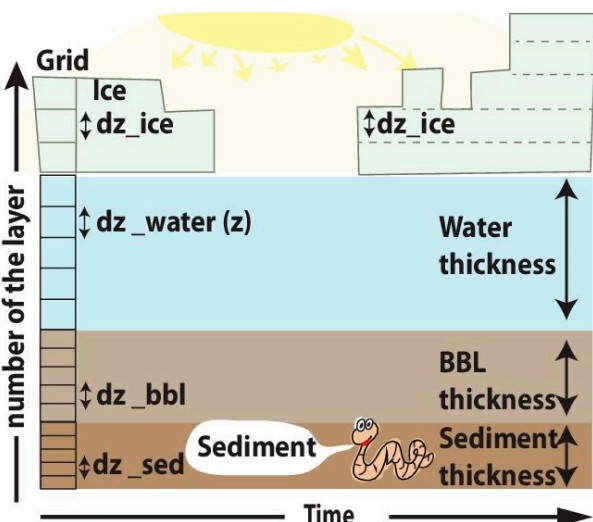

**Figure 1.** The sympagic–pelagic–benthic transport model (SPBM) grid structure.

### 2.3. Irradiance Formulation

FABM biogeochemical models generally need to know the photosynthetically active radiation (PAR), e.g., (mol photons $m^{-2}$ $day^{-1}$), in each layer of the model grid. Some FABM models compute water column PAR given only surface PAR, but they do not assume the existence of the ice column and consider all grid points to be located within the water column. SPBM therefore provides the following simple approach to calculate PAR in both ice and water column domains.

PAR on the surface of water or ice $P_s$ can be calculated from the surface shortwave radiative flux $F_{surf}$ (W m$^{-2}$), depending on the solar declination $k_{decl}$ (degrees):

$$k_{decl} = 23.5 \cdot \sin \frac{2\pi JulianDay - 81}{365}$$
$$F_{surf} = I_m \cos \frac{\pi(latitude - k_{decl})}{180}$$
$$P_s = k_f F_{surf}$$

where $I_m$ is the theoretical maximum of 24-h average surface downwelling shortwave irradiance in air, (W m$^{-2}$); $k_f$ is the factor to convert downwelling shortwave irradiance in air to scalar PAR in water, (mol photons day$^{-1}$ W$^{-1}$) [32]. Alternatively, $P_s$ (or $F_{surf}$) can be read from an input file.

In the presence of ice, PAR after considering albedo influence $P_a$ becomes [33]:

$$\text{if snow depth} \leq 5 \text{ mm} : P_a = P_s k_{scatter}(1 - A_{ice})$$
$$\text{if snow depth} > 5 \text{ mm} : P_a = P_s k_{scatter}(1 - A_{snow})e^{-k_{snow}z_{snow}}$$

where $k_{scatter}$ is the fraction of radiation transmitted through the highly scattering surface of the ice, dimensionless; $A_{ice}$ is the ice albedo for visible light, dimensionless; $A_{snow}$ is the snow albedo for visible light, dimensionless; $k_{snow}$ is the snow light extinction coefficient, (m$^{-1}$); $z_{snow}$ is the snow depth, (m).

PAR at any depth in the ice $P(z_{ice})$ is given by:

$$P(z_{ice}) = P_a e^{-k_{ice}z_{ice}}$$

where $k_{ice}$ is the ice light extinction coefficient, (m$^{-1}$); $z_{ice}$ is the ice depth, (m).

PAR in the water column $P(z_{water})$ is calculated according to the Beer–Lambert formulation:

$$\text{if there is ice} : P(z_{water}) = P(z_{IceBottom})e^{\int_0^z K_{water}(z_{water})dz_{water}}$$
$$\text{if there is no ice} : P(z_{water}) = P_s e^{\int_0^z K_{water}(z_{water})dz_{water}}$$

where $P(z_{IceBottom})$ is the PAR at the ice bottom layer; $K_{water}$ is the vertically varying water light extinction coefficient provided by the FABM models, describing attenuation due to living and non-living optically-active substances, (m$^{-1}$); $z_{water}$ is the water layer depth, (m).

In the sediment domain, PAR equals zero in all layers.

## 2.4. Initial and Boundary Conditions; the Forcing Data

Initial conditions for all state variable concentrations are provided through FABM using its YAML type configuration file [20]. By default, zero gradient boundary conditions are used at upper and lower boundaries for all state variables except $O_2$ and $CO_2$. Diffusive fluxes of $O_2$ and $CO_2$ are provided by the biogeochemical model through FABM at the surface boundary (only for ice-free periods) and are set to zero at the lower boundary. It is possible to change both boundary conditions according to the user's needs.

SPBM requires time-dependent input forcing for the entire period of simulation for the water column (turbulent diffusivity (m$^2$ s$^{-1}$) on layer interfaces; temperature (C) and salinity (psu) on layer centers) and for the ice column (total thickness (m), snow thickness (m), and surface temperature (C)). Additional forcings may be required depending on the FABM biogeochemical models. Downwelling shortwave radiation and PAR can be read from an input file instead of using the formulae provided in Section 2.3. Other optional input forcing includes brine volumes and diffusion

coefficients in the ice, input fluxes at the water surface, and horizontal mixing fluxes at any depth. Input fluxes are based on concentrations C which can be provided in three ways: read from text or NetCDF file; set as fixed sinusoidal variation in time defined by a maximum value M and Phase parameters ($C = 2^{-1}M + 4^{-1}M(1 + \sin(365^{-1} \cdot 2\pi(\text{JulianDay} - \text{Phase}))))$; set as fixed constant value. M and the boundary concentrations C should be in units corresponding to the state variables of the appropriate FABM model, Phase is in (days). SPBM uses input data files in NetCDF and text formats.

## 3. Results—Test Runs

The purpose of the test runs is only to demonstrate the flexibility of SPBM and its relevance to Arctic marine modeling. Rigorous, site-specific adaptation and skill assessment of particular SPBM- 'biogeochemical model' configurations are not within our present remit. SPBM itself does not require validation since it is based on a standard advection–diffusion solver (with a possibility to solve within the ice, water, and sediment domains simultaneously). However, mass conservation of state variables has been checked.

The test runs use forcing data from a regional oceanic modeling system (ROMS) simulation to provide a hydrodynamic scenario for the Laptev Sea. The whole period of this simulation spans a period from 1980 till 2010. A time-dependent total ice thickness and a time-independent water column structure were derived from this simulation, while the BBL was inserted with the following parameters: width = 50 cm, resolution = 10 cm. The grid in the sediment domain was continued for another 10 cm with resolution 2 cm.

For the test simulations, we use FABM to combine components from two published biogeochemical models. Here we will explain only the most basic aspects of these models; the reader can find detailed descriptions in the provided references. We remind that SPBM calculates only the transport terms in Equation (1), while the FABM biogeochemical modules provide the combined sources-minus-sinks terms $R_i$ and the sinking velocities u in the water column. FABM model formulations and parameter values were derived from existing parameterizations with some limited adaptation to the Arctic scenario.

The first biogeochemical model is the European regional seas ecosystem model, ERSEM [21]. Originally a coastal ecosystem model for the North Sea, ERSEM has evolved into a generic tool for ecosystem simulations from shelf seas to the global ocean. Model dynamics within each functional group describe processes occurring inside a 'standard organism' [34,35]. ERSEM accounts for flexible elemental stoichiometry in planktonic processes by allowing decoupled fluxes of carbon, nitrogen, phosphorus, silicate, and chlorophyll a. This requires multiple state variables to describe each functional group biomass (e.g., diatom carbon, diatom nitrogen, etc.) and results in a relatively complex model.

The second biogeochemical model is the bottom-redox model (BROM) biogeochemistry module [22]. This model represents key biogeochemical processes in the water and upper sediments, with a focus on oxygen dynamics and redox biogeochemistry. Compared to ERSEM, it simulates the coupled cycles of more elements (N, P, Si, C, O, S, Mn, and Fe), resolves more structure in the bacterial community (four functional groups vs. one in ERSEM), and calculates carbonate chemistry in more detail; however it assumes fixed stoichiometry for all forms of organic matter (nitrogen currency), resolves only one functional group each for phytoplankton and zooplankton, and does not resolve dissolved organic matter into different lability classes (in the present version).

The general coupling scheme is illustrated in Figure 2. A quasi-stationary solution is derived from two spin-ups, repeating the first day 100 times and then repeating the first year 10 times. Along with SPBM requirements there is additional forcing required by ERSEM and BROM: wind speed (m s$^{-1}$) and concentration of $CO_2$ in air (ppm). Surface shortwave radiation at sea-surface level is provided by ROMS and is read from an input file.

We present two test cases with the same hydrophysical forcing but different FABM model configurations. The first demonstrates the simulation of multiple primary producer functional groups and shows their variability in contrasting conditions. The second test case demonstrates changing redox conditions in the three domains in response to a constant input of organic matter (OM). The main

joint parameter values and forcing properties are provided in Appendix C (common parameters in Table A1, ice parameters in Table A2, sediments parameters in Table A3, irradiance parameters are provided in Table A4, and forcing properties in Table A5).

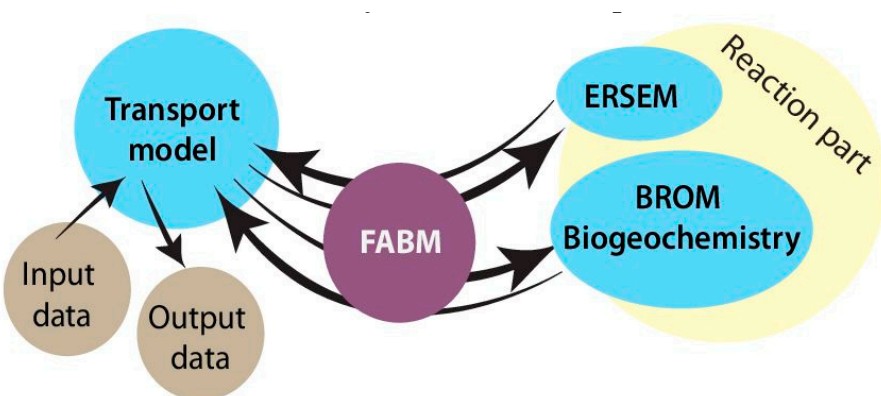

**Figure 2.** The general scheme of models coupling.

### 3.1. Test Case 1

According to the [36], there are few tools yet developed to simulate different groups of ice algae. Therefore, we constructed a test case to simulate different primary producers in different ice conditions. We used the ERSEM primary producer functional group formulation with parameterizations from [21] to simulate diatoms, nanophytoplankton, picophytoplankton, and microphytoplankton. One more algal group called ice diatoms was added, based on an original diatom primary producer parametrization which we adapted to improve growth in low irradiance conditions (see Appendix D). Thus, in total we used five groups. All groups were given the same initial conditions (prior to spin-up) in both ice, water, and sediment domains; hence differences in the steady state abundances were determined by the environment and the growth parameters/sinking velocities of the different functional groups. To calculate pH a corresponding module from BROM was connected. All configuration files are available in the supplementary material.

Figure 3 shows a SPBM–ERSEM–BROM coupled system output (for chlorophyll a, pH, oxygen, and nutrients) in the ice and upper water column layers during the period with maximum ice algae chlorophyll a concentration. This maximum is a result of thin ice (<50 cm) during at least one month and favorable irradiance conditions. Chlorophyll a, oxygen, and nutrients are all state variables and therefore output as concentrations per unit total volume; pH is a diagnostic variable and is output as the negative logarithm of hydrogen ion concentrations of brine or seawater. The modeled values were compared with observed concentrations of biogeochemical tracers in sea ice during spring in the Arctic provided by [8] (p. 210). Most model ranges fall inside the observational ranges (see Table 1). Also, Figure 3 shows that the modeled vertical distributions in sea ice reproduce some commonly observed features [8]: during ice melt chlorophyll a concentrations are highest in the bottom layer; during freezing the pH increases in the upper ice layers, reaching values higher than 10 during winter (see supplementary material) in accordance with observations [37]; nutrients have maximum values on the lowest ice layer, phosphates and nitrates are almost depleted in all ice layers except the bottom one; oxygen profiles in ice have a complicated structure with two maxima, in the middle and the top ice layers. Modeled oxygen are somewhat low compared to the observational range in [8,38] (see Table 1). Observed values include oxygen from gas bubbles that are incorporated into the ice and which are not available for biogeochemical reactions in brine channels, while the modeled ice oxygen is only the oxygen dissolved in ice brines.

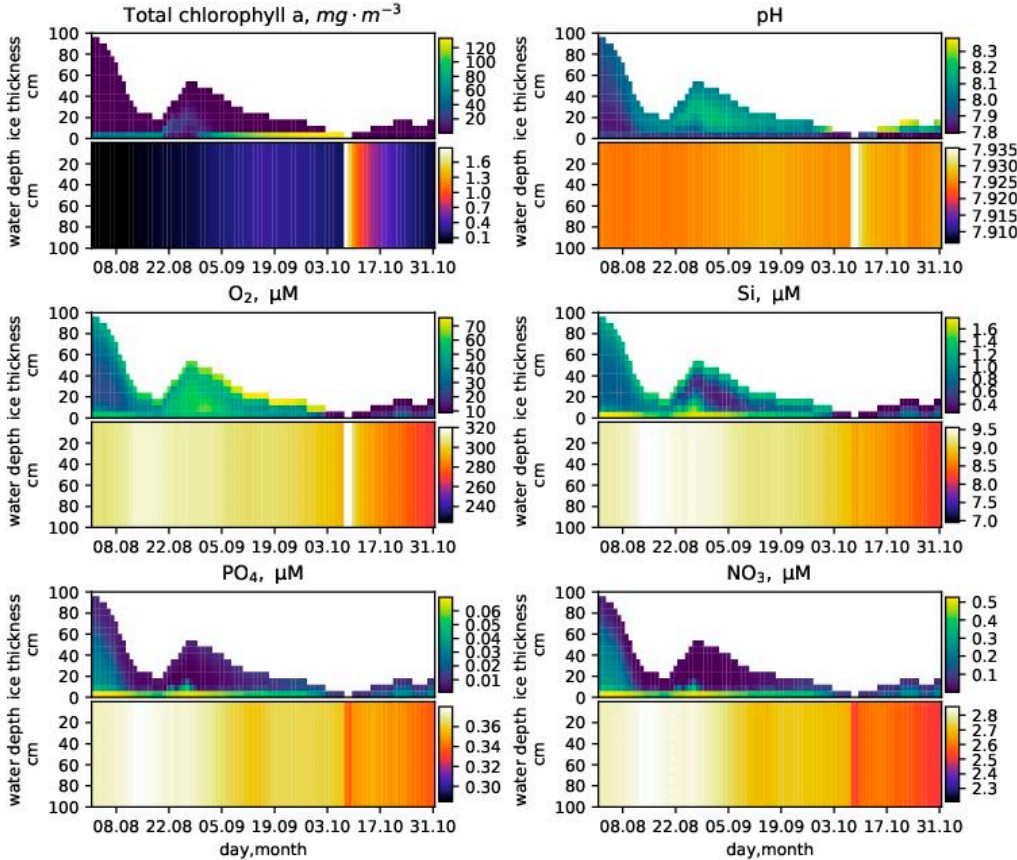

**Figure 3.** Variability of total chlorophyll a, pH (total scale), oxygen, silicon, phosphate, and nitrate in the ice (upper part of respective graphs) and upper water column layers (lower part) during the period with maximum ice algae chlorophyll a concentrations (SPBM–European regional seas ecosystem model (ERSEM)–bottom-redox model (BROM) coupled system output).

**Table 1.** Comparing SPBM–ERSEM–BROM output during ice melting period with concentrations of biogeochemical tracers in sea ice and surface seawater observed for typical spring conditions in the Arctic according to [8] (p. 210). For the observed values in water no value range is provided by the authors.

| Parameter | Si µM | PO$_4$ µM | NO$_3$ µM | Chl a mg m$^{-3}$ | O$_2$ µM |
|---|---|---|---|---|---|
| Observed in ice | - | 0–0.7 | 0–1 | 1–100 | 50–250 |
| Modeled in ice | 0.4–1.6 | 0.01–0.07 | 0–0.5 | 1–120 | 50–80 |
| Observed in water | - | ≈1.2 | ≈7 | ≈1 | ≈380 |
| Modeled in water | 8.5–9.5 | 0.34–0.38 | 2.5–2.9 | 0.1–0.7 | 300–320 |

Figure 4 shows ice/water biomass concentrations of the five primary producer functional groups during the year of maximum primary production and the preceding year. The year of maximum primary production (1983) shows the springtime migration of the modeled ice diatoms and diatoms from the top to the bottom ice layer. The nanophytoplankton and picophytoplankton remain in the uppermost ice layers, while the microphytoplankton migrate downwards during the final stage of the melting season. All modeled primary producers start growing from the surface ice layers, where they were frozen during previous year ice buildup. In years with high irradiance and low summertime ice cover the concentrations of nanophytoplankton and picophytoplankton in the water column are much higher (see supplementary material) and can exceed the diatom concentration.

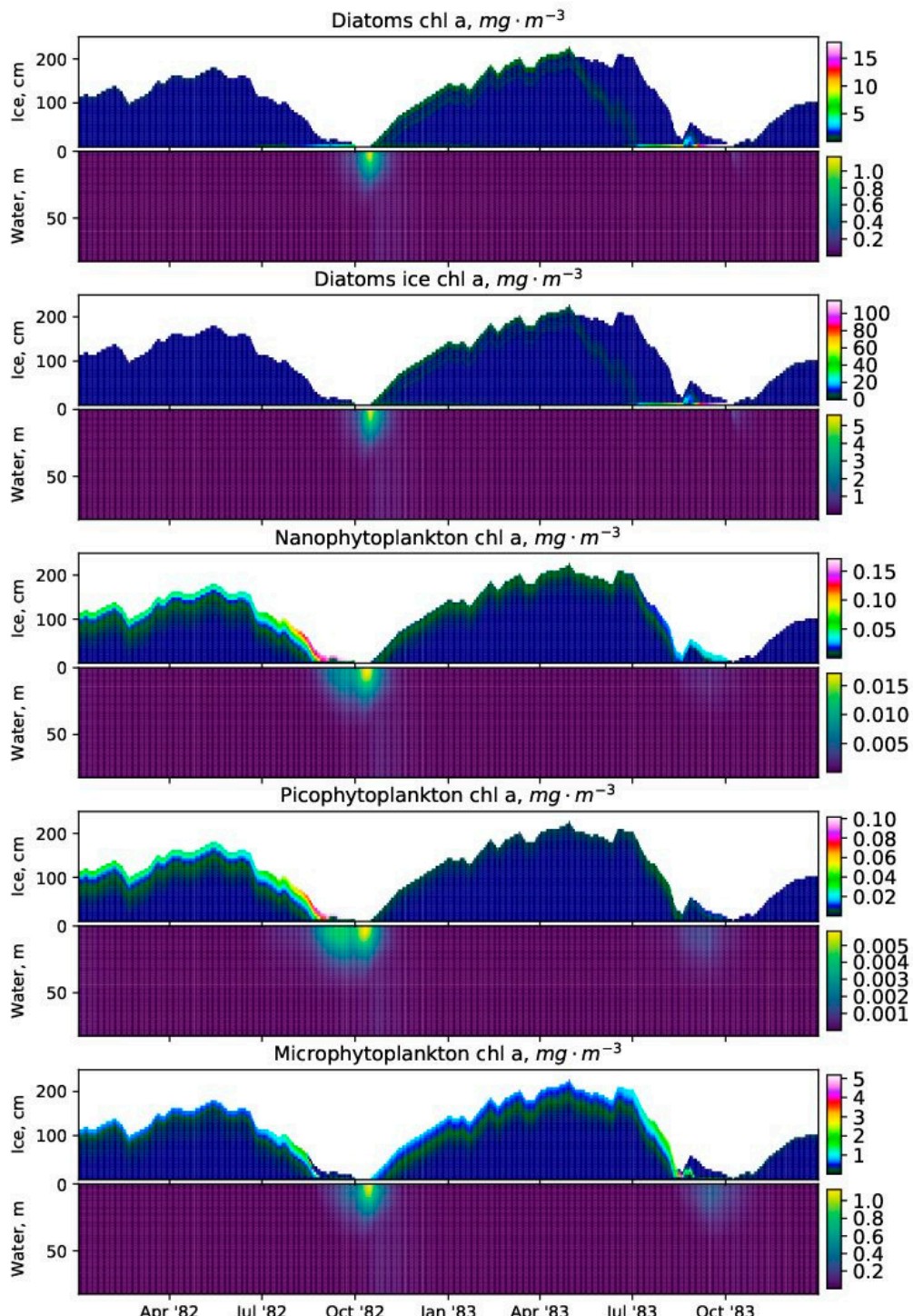

**Figure 4.** Variability of ERSEM primary producer functional groups during the maximum production year (1983) and its preceding year: diatoms chl a, ice diatoms chl a, nanophytoplankton chl a, picophytoplankton chl a, and microphytoplankton chl a (SPBM–ERSEM–BROM coupled system output).

*3.2. Test Case 2*

Test case 2 demonstrates the potential utility of SPBM for studying anaerobic processes in the sea ice, water, and sediment columns. Here we use BROM to simulate biogeochemical processes occurring in low oxygen environments. The BROM configuration file is provided following the link in the code availability section. To facilitate suboxic conditions, we forced a supplementary flux of particulate

OM to the upper level of the water column (see along with other forcing properties in Table A5). For presentation, we chose the same years as for test case 1. Figures 5–7 show some of the available BROM variables in the ice, water, and sediment domains. Concentrations in ice are strongly driven by surface water concentrations, which in turn are strongly influenced by hydrophysical conditions. For variables not involved in redox reactions, vertical distributions in the ice column mainly reflect temporal distributions in the upper water layer during freezing.

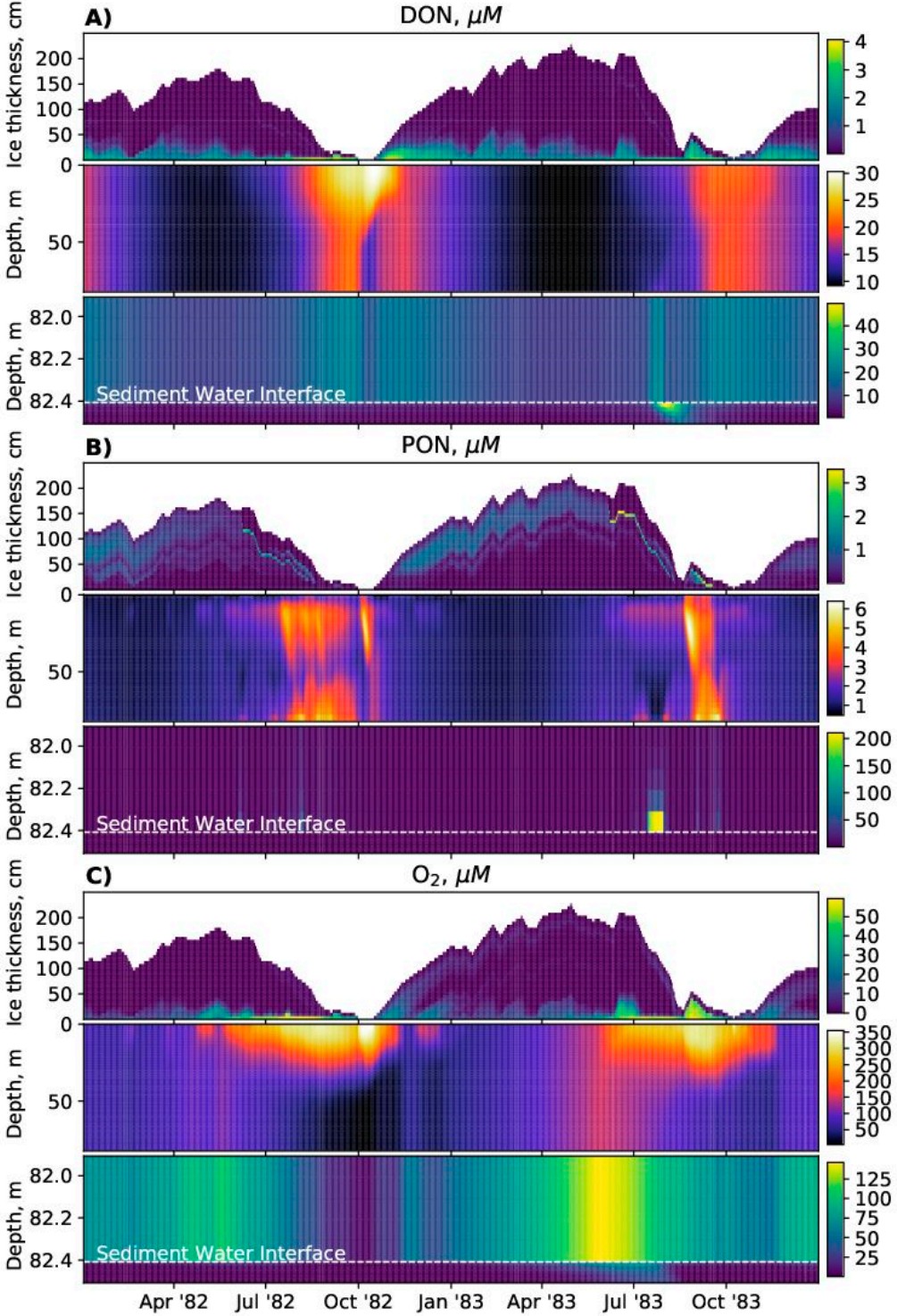

**Figure 5.** Variability of dissolved organic matter (OM) (DON) (**A**), particulate OM (PON) (**B**) (both in nitrogen units), and $O_2$ (**C**) in suboxic conditions (SPBM–BROM coupled system output).

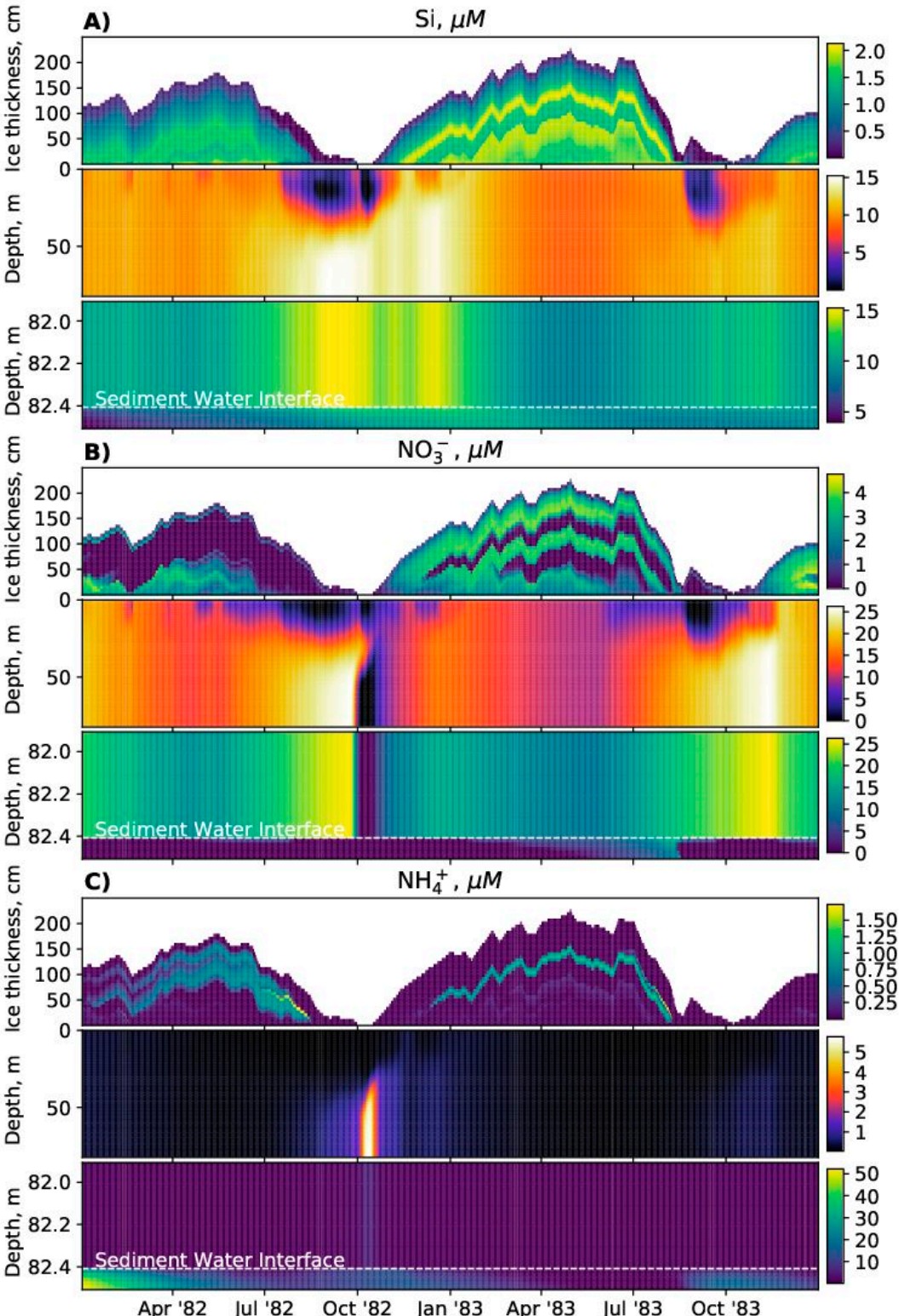

**Figure 6.** Seasonality of Si (**A**), $NO_3^-$ (**B**), and $NH_4^+$ (**C**) concentrations in suboxic conditions for 1982 and 1983 (SPBM–BROM coupled system output).

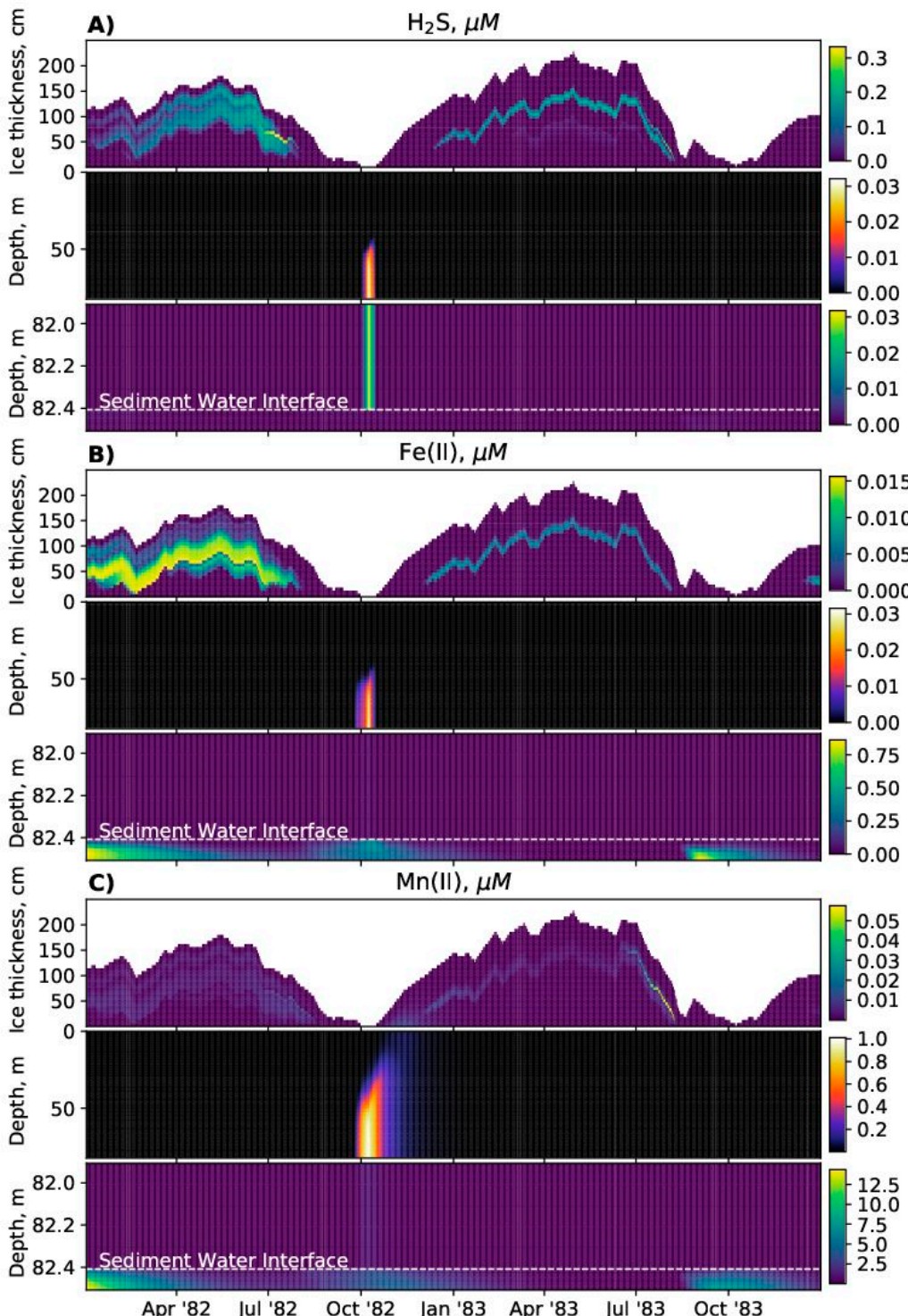

**Figure 7.** Seasonality of H$_2$S (**A**), Fe(II) (**B**), and Mn(II) (**C**) concentrations in suboxic conditions for 1982 and 1983 (SPBM–BROM coupled system output).

The simulation reproduces some general features of sea ice–water–sediment seasonal biogeochemistry connected with the seasonal production and decomposition of OM. Phytoplankton (one state variable in BROM) starts to bloom in the ice and below the ice during ice melting (see supplementary material). Phytoplankton blooms in the upper water column lead to a seasonal increase in dissolved organic matter (DON, Figure 5A, in nitrogen units), dead particulate organic matter (PON, Figure 5B, in nitrogen units), and dissolved oxygen (Figure 5C), followed by oxygen consumption in

the lower water column/BBL and the generation of reduced forms of nitrogen ($NO_2^-$, $NH_4^+$) at depth (Figure 6C). DON is incorporated into the ice during freezing, but since BROM only models the labile fraction at present, the model DON is rapidly oxidized and contributes little to OM in the upper ice layers (Figure 5A). Here, the main reduction agent is PON, which occurs in ice due to decomposition of zooplankton (Figure 5B). The modeled oxygen in ice is almost depleted (Figure 5C). In the upper sediments, oxygen is depleted almost year-round, indicating active redox processes in this domain.

In contrast to nitrogen species, silicate does not participate in redox reactions and therefore its distribution in the ice mainly reflects the surface water concentrations during ice buildup, which are mainly driven by phytoplankton uptake and mixing with meltwater and deep water (Figure 6A). Similarly, silicate in the sediments mainly reflects bottom water concentrations, which are mainly driven by remineralization, mixing with surface water, and transport into the sediments. In the case of no additional supply of OM, and in absence of low oxygen conditions, mineral forms of nitrogen (e.g., $NO_3^-$) would have very similar profiles to silicate. As it is, the low wintertime concentrations of oxygen in the surface water (Figure 5C), and in the latter case bacteria can use nitrate as an oxidizing agent. Therefore, during ice freezing in some ice layers (where there is more oxygen) nitrification occurs and within other layers (where there is less oxygen) denitrification and anammox processes prevail (see supplementary material). By the start of the melting season there are almost no active nitrogen redox transformations in the ice (see supplementary material), and the distributions of nitrates, nitrites, and ammonium in the ice are mainly determined by melting processes.

In Figures 6 and 7 it is demonstrated that a long ice melting season leads to a strong stratification in the water column that prevents oxygen supply from the surface layer downwards. OM produced during the phytoplankton bloom leads to oxygen consumption in the subsurface layers, thereby triggering the process of denitrification and the consumption of nitrate and nitrite for OM decomposition. Variability of nitrogen species illustrates this as a temporal decrease of concentrations of nitrate and an increase of ammonia in the water column (Figure 6). Before and after this nitrate minimum, nitrate maxima are formed. For this oxygen depleted period the model also predicts an increase in content of dissolved $Mn^{2+}$ and dissolved $Fe^{2+}$ (Figure 7B,C), connected with the reduction of oxidized forms of these metals. Finally, trace concentrations of hydrogen sulfide appear in the bottom water (Figure 7A) due to the process of sulphate reduction that starts when oxidized forms of nitrogen, manganese, and iron are depleted. This sequence of changes of electron acceptors corresponds to the theory [39] and to the classical features of the structure of the water column redox interfaces, e.g., in the Black Sea or the Baltic Sea [40–42]. Similar temporal changes are observed in places with variable redox conditions, e.g., Norwegian fjords and coastal lagoons [43–45]. The duration and intensity of oxygen depletion in the water column (and therefore in the sediments) are determined by the oxygen and OM supply in combination with the peculiarity of the ice regime (time and duration of the ice-melting period), that affect the distributions of chemical and biological parameters in the water column.

## 4. Discussion

Overall, the test simulations demonstrate potentially important interactions between ice, water, and sediment biogeochemistry, as well as how distinct vertical structure can emerge in the ice (Figures 3–7) and in the sediments (Figures 5–7). This suggests that SPBM is a potentially useful tool for marine biogeochemical modeling in the Arctic. SPBM is not restricted to a particular biogeochemical model. Instead, it can calculate the transport of variables provided by any model (or models) already available in the FABM library or written according to the FABM application programming interface. SPBM has been developed initially to study vertical interactions between ice, water column, and sediments. However, by setting the number of ice or sediment layers to zero a user can choose domains of interest. Also, SPBM can be applied to test newly developed biogeochemical models since the user can vary the SPBM grid from a box to a multilayer model. Compared to 3D models, 1D tools are less suitable for forecasts but can be used as "complex calculators" for processes investigations. In this regard, SPBM provides an important ability to quantify fluxes of biogeochemical elements

between the ice, water, and sediment domains. SPBM thus contains all necessary domains and is an ideally suited instrument for studying polar biogeochemistry especially in shallow waters. However, there are some important limitations of the present version that the user should consider.

First, as a 1D model, SPBM does not explicitly account for horizontal transports (advection/diffusion) which may be significant in the water column. Horizontal mixing fluxes can however be implemented, with depth/time-varying mixing concentrations and mixing rate coefficients. Of course, this latter is more likely to give reasonable results if the real horizontal transports are of a mixing/exchange character, rather than e.g., a persistent advective flux divergence. An alternative that could be implemented in future versions is to allow arbitrary depth/time-varying horizontal transport contributions or advective timescales to be read from file, perhaps based on a 3D biogeochemical model simulation e.g., [46].

Second, the present ice parameterization in SPBM is most suitable for one-year-old sea ice since it is largely based on formulations from [26]. If this is not adequate, the ice brine volume and diffusion coefficients can alternatively be taken from an ice thermodynamic model (if available). Also, the present SPBM implementation of gases in ice takes into account only the dissolved part of them and does not include bubbles. Since most of the biogeochemistry processes in ice occur in brine channels it is not crucial in the context of oxygen availability for redox reaction representation. But the fact that this process is not included can result in overestimation of initial values for dissolved gases incorporated in an ice core (e.g., [38] estimate the bubbles contribute roughly a third of the total oxygen content in ice). This will also be addressed in further work.

A third potential weakness of SPBM is in the parameterization of transport in the sediments (porosity, diffusion, and burial velocities). Equations (2) and (3) assume a fixed (time-invariant) porosity profile, a fixed deep burial velocity for solutes and particulates, and no net contribution of biogeochemical transformations to the total particulate volume fraction (see [22] Appendix B). This in turn implies a fixed total particulate volume flux or "sedimentation rate" at the SWI. Future versions might allow some temporal variability in this total volume flux, perhaps using input files and/or including an explicit contribution from the seasonal sinking flux of SPBM-modeled particulates (e.g., PON). A subtlety with the latter approach is that if, as in SPBM, the bottom layer of the water column is considered a "fluff layer" with particles entering at sinking velocity and leaving at burial velocity, then additional assumptions are required to determine the flux of modeled particulates that enters the sediments and becomes part of the sediment matrix, rather than remaining as "fluff" on the SWI [22]. A related issue here is the lack of explicit erosion/resuspension processes in the present SPBM model. Within the sediments, the neglect of solute adsorption in the present version may also be an issue in some applications.

Fourth: the ability of FABM to combine state variables from different models in a modular fashion (as specified in the configuration file) should accelerate the development of well-parameterized sympagic–pelagic–benthic biogeochemical modules within SPBM, and will also ensure that the same module code is used in any subsequent 3D simulations as long as the 3D model is also FABM-coupled (e.g., ROMS, FVCOM, GETM, MOM, NEMO). However, care is needed to ensure compatibility between state variables, and differences in model structure and currencies may present obstacles. For example, in test case 1 we combined state variables from ERSEM and BROM, seeking to combine the pelagic process resolution in ERSEM with the BROM resolution of redox biogeochemistry and sedimentary nutrient recycling. However, it was not possible to fully couple these models because they use different currencies (BROM uses nitrogen units, while ERSEM uses carbon, nitrogen, phosphorus, and silicon). Complete coupling of ERSEM and BROM may ultimately require some recoding of BROM state variable modules to allow for flexible elemental stoichiometry. Furthermore, while FABM allows modules to be repurposed to describe domain-specific variables (e.g., ice diatoms from the ERSEM primary producer module in test case 1) the user must exercise caution to ensure that parameter values are suitably adapted and that the module has sufficient flexibility to describe the domain-specific variable (if not, a new FABM module may need to be written). Also, a structural approximation that

may be adequate in one domain (e.g., the use of one lability class for DON in BROM) may not be adequate in another (e.g., for DON in ice, as in test case 2).

Finally, the SPBM approach of simulating all variables in all domains implies some computationally inefficiency, e.g., calculating the transport of minute quantities of phytoplankton in the sediments. Future versions could implement domain-specific screening switches to avoid this, although in a 1D context this may not be necessary. Also, a base assumption that "everything is everywhere, but the environment selects" [47] can be insightful. For example, in test case 1, SPBM simulated some limited growth of phytoplankton groups in the ice domain, and this may be important in seeding the phytoplankton blooms in the water column following ice melt [48].

## 5. Conclusions

We aimed to develop a flexible 1D vertical transport model to allow simultaneous simulation of the marine biogeochemistry of 3 different media: ice, water, and sediments. The resulting sympagic–pelagic–benthic model (SPBM) includes vertically-resolved ice and sediment domains, and allows fine resolution of the benthic boundary layer. SPBM reads input file data on ice growth and water column physics (and optionally also brine volumes and ice diffusivity) and uses the Framework for Aquatic Biogeochemical Models (FABM) to provide a user-defined model for biogeochemical transformations and water column sinking velocities, based on published models in the FABM library and possible new modules written by the user. Two test simulations demonstrated the potential utility of SPBM for modeling systems with strong interactions between ice, water column, and sediment biogeochemistry, as are often found in the Arctic Ocean and shelf seas. In the first test case, the FABM coupling was used to combine modules from two complex biogeochemical models (ERSEM and BROM) and to adapt an existing ERSEM diatom parameterization to simulate a new sea ice diatom group in combination with the other four ERSEM phytoplankton groups. The simulation demonstrated a strong interaction between water column and ice domains with respect to algal blooms, with the ice providing seed populations of phytoplankton and the water column providing an income of nutrients. It also demonstrated that different groups of primary producers have different spatial and temporal variabilities both in the ice and water domains due to different requirements and limitations. A second test case demonstrated strong interactions between ice, water, and sediment domains, with spatial variability of nutrients in sea water during sea ice congelation season determining the processes occurring in the ice core in the following winter, and the melting season features determining the redox reactions occurring in the sediments. Although there are some notable limitations of the present SPBM version, the results herein suggest that SPBM can already provide a useful tool for tuning existing biogeochemical models, accelerating the development of new biogeochemical models for regions with strong interactions between ice, water column, and sediment, and for investigating the potential importance of such interactions in determining the response of Arctic ecosystems to local and global anthropogenic drivers.

**Supplementary Materials:** The code is available online at https://github.com/BottomRedoxModel/SPBM, (git tag v0.2), supplementary pictures are available online at http://www.mdpi.com/2073-4441/11/8/1582/s1.

**Author Contributions:** Conceptualization, S.Y.; methodology, S.Y., P.W.; software, S.Y., E.P.; validation, S.Y.; visualization, E.P.; resources, H.B.; writing—original draft preparation, S.Y.; writing—review and editing, S.Y., P.W., E.P., E.Y., S.P., H.B.

**Funding:** E.P., P.W. and E.Y. were supported by the Norwegian Foreign Ministry and its Arctic 2030 program (project PERMAFLUX), the FRAM High North Research Centre for Climate and the Environment under the Ocean Acidification Flagship, and Norwegian Research Council project no. 272749 ('Aquatic Modeling Tools', SkatteFUNN).

**Conflicts of Interest:** The authors declare no conflict of interest.

## Appendix A

*Porosity* $\varphi(z)$ at depth z in the ice column is considered as relative volume of brine channels in ice [26]:

$$\varphi(z) = \frac{\rho_i(z)S_i(z)}{\rho_b(z)S_b(z)}$$

*Brine salinity*, $S_b(z)$ (ppt) [26] and corresponding sea ice temperature (degrees Celsius), $T_i(z)$:

$$S_b(z) = \alpha_0 + \alpha_1 T_i(z) + \alpha_2 T_i(z)^2 + \alpha_3 T_i(z)^3$$

$$T_i(z) = \text{AirTemperature} + \frac{(\text{WaterTemperature} - \text{AirTemperature})}{\text{IceThickness}} z$$

where $\alpha_0$, $\alpha_1$, $\alpha_2$ and $\alpha_3$ are different for 3 ranges of temperatures:

| $T_i$ | $\alpha_0$ | $\alpha_1$ | $\alpha_2$ | $\alpha_3$ |
|---|---|---|---|---|
| $-1.85 > T_i \geq -22.9$ | −3.9921 | −22.700 | −1.0015 | −0.019956 |
| $-22.9 > T_i \geq -44$ | 206.24 | −1.8907 | −0.060868 | −0.0010247 |
| $-44 > T_i \geq -54$ | −4442.1 | −277.86 | −5.501 | −0.03669 |

*Sea ice salinity*, $S_i(z)$ (ppt) [10,49]:

$$S_i(z) = 19.539 \cdot Z_p^2 - 19.93 \cdot Z_p + 8.913$$

where $Z_p$ is the ratio between the distance from the ice surface and ice thickness.

*Brine density*, $\rho_b(z)$ (g m$^{-3}$) [50]:

$$\rho_b(z) = (1 + cS_b(z)) \cdot 10^6$$

where $= 8 \times 10^{-4}$ g m$^{-3}$ ppt$^{-1}$.

*Sea ice density*, $\rho_i(z)$ (g m$^{-3}$) [26]:

$$\rho_i(z) = \frac{\rho_0 \rho_b(z) S_b(z)}{\rho_b(z)S_b(z) - S_i(z)(\rho_b(z) - \rho_0)}$$

where $\rho_0 = 912 \times 10^3$ g m$^{-3}$ is the density of pure ice.

## Appendix B

The molecular diffusivity $D_m$ (m$^2$ s$^{-1}$) mixes concentrations of the solutes in units (mmol m$^{-3}$ solutes). While the bioturbation diffusivity $D_b$ (m$^2$ s$^{-1}$) mixes concentration of the both solutes and solids in units (mmol m$^{-3}$ total volume). So there is a flux for solutes on the SWI:

$$\begin{aligned}
F_{swi} &= -\varphi_{swi} D_m \frac{\frac{C_a}{\varphi_a} - \frac{C_b}{\varphi_b}}{\Delta z} - D_b \frac{C_a - C_b}{\Delta z} = \\
&= \frac{C_a\left(-\frac{\varphi_{swi}}{\varphi_a}D_m - D_b\right)}{\Delta z} + \frac{C_b\left(\frac{\varphi_{swi}}{\varphi_b}D_m + D_b\right)}{\Delta z}
\end{aligned} \tag{A1}$$

In the 1D model the flux is calculated in the form where the porosity factor $P_f(z_{a,b})$ should be determined:

$$\begin{aligned}
F_{swi} &= -\varphi_{swi}(D_m + D_b)\frac{P_f(z_a)C_a - P_f(z_b)C_b}{\Delta z} = \\
&= -\frac{\varphi_{swi}(D_m + D_b)P_f(z_a)C_a}{\Delta z} + \frac{\varphi_{swi}(D_m + D_b)P_f(z_b)C_b}{\Delta z}
\end{aligned} \tag{A2}$$

Comparing Equation (A1) and Equation (A2):

$$P_f(z_{a,b}) = \frac{\frac{\varphi_{swi}}{\varphi_{a,b}}D_m + D_b}{\varphi_{swi}(D_m + D_b)}$$

For solids since $D_m = 0$ and $1 - \varphi_{swi}$ instead of $\varphi_{swi}$:

$$P_f(z_{a,b}) = \frac{1}{1 - \varphi_{swi}}$$

where C is the concentration of the variable, (mmol m$^{-3}$ total volume); $\varphi$ is porosity, dimensionless. Subscripts a, b, and swi determine a location of the corresponding variables: a means the layer above, b—the layer below, swi—on the SWI.

## Appendix C

**Table A1.** Common parameters.

| Parameter | Description | Value | Unit | Reference |
|---|---|---|---|---|
| t | Time step | 300 | s | |
| $D_0$ | Infinite-dilution molecular diffusivity | $10^{-9}$ | m$^2$s$^{-1}$ | [28] |
| $V_{ws}$ | Wind speed | 5 | ms$^{-1}$ | |
| $CO_2(g)$ | Concentration of $CO_2$ in air | 380 | ppm | |

**Table A2.** Ice parameters.

| Parameter | Description | Value | Unit | Reference |
|---|---|---|---|---|
| $D_m(s)$ | Diffusivity on sea-water interface | $10^{-5}$ | m$^2$s$^{-1}$ | [51] |
| $u_d$ | Diatoms vertical movement velocity | 3 | cm d$^{-1}$ | |
| $F_{vb}$ | Flux rate from the brine channels | $10^{-8}$ | ms$^{-1}$ | [26] |
| $z_s$ | Thickness of the ice layer | 0.06 | m | |
| $\varphi_{min}$ | Minimum porosity to enable brine convection | 0.072 (0.12) | - | [26] |

**Table A3.** Sediments parameters.

| Parameter | Description | Value | Unit | Reference |
|---|---|---|---|---|
| $\varphi(z_\infty)$ | Porosity at the infinite sediments depth | 0.8 | - | [27] |
| $\varphi(z_0)$ | Porosity at the sediments–water interface | 0.95 | - | [27] |
| $k_\varphi$ | Coefficient for exponential porosity change | 0.04 | m | [27] |
| $\mu_d$ | Relative dynamic viscosity | 0.94 | - | [28] |
| $K_{O2}$ | Oxygen half-saturation constant | 5 | mmol m$^{-3}$ | [22] |
| $z_{cb}$ | Constant bioturbation activity layer width | 0.02 | m | [28] |
| $D_{bm}$ | Maximum bioturbation diffusivity | $10^{-11}$ | m$^2$s$^{-1}$ | [28] |
| $F_d$ | Bioturbation decay scale | 0.01 | m | [28] |
| $u_b$ | Deep burial velocity | $10^{-10}$ | ms$^{-1}$ | [28] |

**Table A4.** Irradiance parameters.

| Parameter | Description | Value | Unit | Reference |
|---|---|---|---|---|
| $k_f$ | Factor converting irradiance to PAR | 0.5 | mol photons d$^{-1}$W$^{-1}$ | [32] |
| $k_{scatter}$ | Fraction of transmitted radiation | 0.97 | - | [33] |
| $A_{ice}$ | Ice albedo | 0.744 | - | [33] |
| $A_{snow}$ | Snow albedo | 0.9 | - | [52] |
| $k_{snow}$ | Snow light extinction coefficient | 4.3 | m$^{-1}$ | [52] |
| $k_{ice}$ | Ice light extinction coefficient | 0.93 | m$^{-1}$ | [33] |

**Table A5.** Forcing properties were chosen according to the values provided by [53].

| Parameter | Position | Type | Value |
|---|---|---|---|
| DIC | Water surface layer | Constant | 1930 mmol m$^{-3}$ total volume |
| TA | Water surface layer | Constant | 2000 mmol m$^{-3}$ total volume |
| PO$_4$ | Water surface layer | Sinusoidal | *M* = 0.4, *Phase* = 130 |
| NO$_3$ | Water surface layer | Sinusoidal | *M* = 3, *Phase* = 130 |
| Si | Water surface layer | Sinusoidal | *M* = 10, *Phase* = 130 |
| O$_2$ | Water surface layer | Sinusoidal | *M* = 330, *Phase* = 130 |
| DON | Water surface layer | Sinusoidal | *M* = 12.5, *Phase* = 130 |
| DIC | Water bottom layer | Constant | 2280 mmol m$^{-3}$ total volume |
| TA | Water bottom layer | Constant | 2350 mmol m$^{-3}$ total volume |

**Appendix D**

**Table A6.** ERSEM photosynthesis parameters.

| Parameter | Diatoms | Ice Diatoms |
|---|---|---|
| g | 1.375 | 1.210 |
| α | 4 | 5.98 |
| β | 0.07 | 0.2 |

In ERSEM there are 4 adjustable parameters that influence gross production without affecting nutrient or temperature limitation: maximum specific productivity at reference temperature (g), initial slope of PI-curve (α), photoinhibition parameter (β), and maximum effective chlorophyll to carbon photosynthesis ratio (φ). Tuning these parameters one can get different photosynthesis–irradiance curves which would represent different irradiance requirements [21] (Figure 13, p. 1333). We wanted our new primary producer functional group to be more tolerant to lower PAR conditions, but we did not want to change significantly its behavior. So, we adjusted only g, α, and β in a way that increased the initial slope of the photosynthesis–irradiance curve but preserved the area under this curve from 0 to 20 Wm$^{-2}$. The parameters for the original ERSEM diatom and the derived parameters for the new ice diatom are presented in Table A6.

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
