# Peer review of "A 1-Dimensional Sympagic–Pelagic–Benthic Transport Model (SPBM): Coupled Simulation of Ice, Water Column, and Sediment Biogeochemistry, Suitable for Arctic Applications"

_water, doi:10.3390/w11081582_

Round 1

Reviewer 1 Report

The manuscript presents a new 1D model to simulate coupled ice, water and sediment biogeochemistry as a plug-in to the FABM framework, using existing models ERSEM and BROM to handle biogeochemical processes.

The paper is very well-written and lays out the model equations and logic clearly, and discusses limitations as well as advantages of the model appropriately. The model is demonstrated in application to two examples. Neither example is a complete case-study comparing model results with field observations, however I feel that is acceptable for the initial presentation of a new model. The paper presents a worthwhile novel contribution to the literature.

The model equations and representation of relevant processes are sound as far as I can tell (bearing in mind that I do not know a lot about ice and snow).

A few points to clarify:

- Is the software available to others? If so, from where?

- Are all ERSEM biogeochemical processes run in ice layers? If so, are the altered boundary layers for diffusion to and across cell surfaces taken into account?

- Nice worm in Figure 1 :) Is the artwork yours or public domain?

A couple of potential avenues for future work (I am not suggesting that these need to be addressed now):

- Does the model represent the effects of freezing on process rates after substances are re-released from ice into the water? Freezing can have a range of effects, including lysis of (phyto- zoo- and bacterio-plankton) cells and break-up of organic matter aggregates.

- It would be nice in future work to see some attention given to process-driven representation of variations in burial velocity and bioturbation

Reviewer 2 Report

The manuscript by Yakubov et al. purports to aim “to develop a flexible and computationally-efficient 1D vertical transport model to allow simultaneous simulation of the marine biogeochemistry of 3 different media: ice, water, and sediments.” I am neither a modeler not a specialist on Arctic or polar biogeochemistry, and my perspective while reviewing this manuscript is that of a marine biogeochemist who specializes on sedimentary organic-matter/nutrient dynamics and redox cycling.

From my viewpoint, therefore, I judge that the authors accomplish their stated aim. Judging from my understanding of biogeochemical models (especially benthic ones) the model structure description appears to be terse but it may be appropriate, given the model design/scheme shown in Figure 2. The documentation of options and appropriate references to the operation of the interconnected models is adequate. I appreciate the discussion of the justification, functionality and limitations of the model; they lay those aspects out clearly. The comparison with actual data is key and the results are presented efficiently in a manner that highlights the potential contributions of this work.

I, therefore, recommend its publication after a few minor considerations:

 “Anoxic conditions”: Test case 2 focuses      on the utility of the model under subaerobic/suboxic or even anoxic      conditions. The authors have to be more cautious with the terminology they      use. The use of “anaerobic” (line 370) and “low oxygen” (line 372) are      appropriate and acceptable in the context they appear in. However, the use      of “anoxic” (which is, generally, accepted to denote [O2]=0) in line 373      and especially in the caption of Figure 5 (line 382) isn’t.      Characteristically, O2 isn’t at 0 for most of the domain displayed in the      figure. I suggest the use of the term “subaerobic” or “suboxic” instead.

P      participation in redox reactions: This is implied in line 399 “In contrast      to nitrogen species and phosphate, silicate does not participate in redox      reactions”. As the authors point out further in this paragraph, nitrate      can be used an electron acceptor in lieu of O2 under suboxic conditions,      and DIN speciation will shift dramatically under variable redox conditions.      However, the participation of phosphate in redox reactions is unheard of.      Phosphate solubility is generally affected by redox conditions, but      phosphate itself is not a reactant but merely the product of organic      matter breakdown, much like silicate (if diatom biomass is decomposing). I      looked for an explanation to this in the rest of the manuscript but didn’t      find one. The easiest remediation is the removal of “and phosphate” from      line 399.

Transport      in the sediments (lines 467-479): Sharp gradients in the sediment column      and the presence of extensive erosion and resuspension may be a pretty      significant problem. Surely, reference to any published studies that      demonstrate the presence or absence of these processes will allow the      reader to understand whether this is indeed a deficiency of the model at      present and, if so, how big. I suspect that if the sediments are      fine-grained, gradients in many redox parameters will indeed be sharp but      constrained very close to the sediment-water-interface, thus allowing the      treatment of the sediment column as a homogeneous box. This is speculation,      obviously. I’d encourage the authors to illuminate the nature of the      problem with the citation of studies that describe the presence/absence      and magnitude of the processes they bring forth in this paragraph.

Solute adsorption      (lines 479-480): This is an issue with key variables such as ammonium, the      dominant DIN species in very hypoxic/anoxic conditions. So, it is a good      point to make here, but I don’t think it’s a prohibitive problem for the      model. It’s one more thing to refine in future versions.

As I close this review, I point out that I was aware of the history of the manuscripts of this study as they are available online through the portal of a publishing house that publicizes the peer-review process. I consulted with MDPI, who confirmed that that previous publication is considered a pre-print, therefore constituting this submission as legitimate. I declare that I purposefully ignored the previously publicized reviews and editorial comments when compiling my own review of this particular manuscript.

Reviewer 3 Report

The manuscript describes a software framework for simulating the marine biogeochemical processes in Arctic regions, which typically consist of ice, water, and sediment domains. Central to this software framework are the biogeochemical reaction network models, the transport model, and the coupling between these. Mainly, in this manuscript, a 1D Sympagic-Pelagic-Benthic transport model (SPBM) is developed and coupled with an existing biogeochemical model (i.e., Framework for Aquatic Biogeochemical Models (FABM)).

While I agree with the general premise of this work, my main concern regarding this manuscript is that the mathematical model is rather poorly motivated.
Basically, from what I understand, the model concept is the following.
There are three zones: porous ice, water column, and porous sediment. Porosity in the ice zone relates to relative volume of brine channels, and the porosity in sediment relates to the relative void volume in the granular matrix. The ice zone consists of two phases, ice and brine; water column consists of only one phase, sea-water; and, sediment zone consists of two phases, granular(soil)-matrix and sea-water. The objective of the model is to track the reactive transport of tracers (e.g., chemical, biological, mineral species) within and across the ice-water-sediment zones as a result of biogeochemical and hydrodynamical processes on geological/seasonal time-scales. These tracers are of two types: solutes and particulates. The solutes exist only in fluid phase (i.e., brine, seawater, porewater) in all three zones, while particulates can exist in all phases and all zones and are transported through different mechanisms.
The main novelty of this work is in the unified description of the reactive transport processes in the three zones. This fact is, however, not adequately motivated. The 1D-transport equation is introduced without first introducing the notions of phases, components (species), porosity, etc. Important details regarding the coupling and numerical solution strategies are also missing.

I also somewhat disagree with the claim that the presented 1D transport model is flexible and computationally efficient, because no basis has been provided for this claim.
Perhaps flexibility can be argued on the basis of modularity of the code. But again, the limitation to 1D is rather restrictive and goes against the notion of flexibility. I think the authors should clearly point out in what aspects the model is flexible.
Computational efficiency is harder to justify because no code performance metrics have been provided in the manuscript.

My overall impression is that this manuscript has scope for publication within this journal, but requires revisions.

Detailed comments

1- The paragraph in lines 94-100 has been introduced without any context or precursor.
Definitions of state variables, diagnostic variables, and dependencies are not clear. By themselves, these terms are much too generic. In general, state variables refer to the independent variables required to uniquely define the state of the system (often, the state variables are synonymous with the primary variables of a system of equations). I presume that the diagnostic variables refer to secondary variables and quantities-of-initerest (e.g. constitutive relationships, physical parameters like diffusivities, etc.) which can be derived from the primary variables at any time step; and again, presumably, dependencies refer to physical/interface/numerical/computational-domain/coupling parameters.

2- What exactly is the FABM module solving? The reaction network for the biogeochemical processes? Please clearly mention the capabilities of FABM and list the relevant outputs/inputs which couple FABM and SPBM codes. (To some extent this information is indeed provided in the manuscript, but its all scattered around (e.g. lines 279-282). It will be helpful to introduce it in the model formulation itself.)

3- In line 100, it is stated that solute and particle concentrations are the state variables, but in line 104, concentrations are referred to as a property of the state variables. Please clarify.

4- In line 105, it is stated that both the solute and particulate concentrations are described in the units of [mmol m^-3 total volume]. Does the total volume refer to the volume of the REV (representative control volume), or that of the respective phases hosting the solute and the particulate components? In marine settings for example, it is conventional to describe solute concentrations in terms of porewater volume and the particle concentrations in terms of dry sediment volume.

5- I am not sure I understand the meaning of lines 118-120. Particulate components embedded in ice-matrix do not diffuse, only those suspended within brine can diffuse?

Some questions regarding the numerics:

6- How is the ice growth modeled? Is it just some time-dependent function/lookup-table? Or is it solved as a Stefan-type problem? How are the grids in the ice-zone adapted?

7- How is the transport eqn. discretized in space?

8- Is the transport equation solved separately for each component or is the full system solved simultaneously?

I am assuming that in the operator splitting scheme, the transport operator is split into a convection-diffusion (CD) part and a reaction (R) part. If the transport eqn. is solved without any reformulation, then the CD part is linear and each CD term can be solved separately, saving a lot of computational time.

9- The coupling in Eqn. 1 appears through the source/sink terms. How is this coupling handled? Please elaborate on your solution strategy for the full system of PDAEs comprising this model. Schematic in Fig. 2 lacks important details about coupling terms and the type of input/output data.

10- How are the equality constraints (due to equilibrium reactions) handled in the reaction network? How are these constraints imposed during the coupling between FABM and SPBM? This is particularly relevant when determining the source/sink terms due to equilibrium reactions, and can have a large effect on the accuracy of the coupling strategy.

11- In the reference (Butenschön2012) the sensitivity of two different operator splitting schemes is analyzed. Which one of these operator splitting schemes is used in this work and how is it realized within this software framework? Please also note that in line 200, it is stated that operator splitting is a time stepping scheme, but that is not completely correct. Operator splitting, in the context of this model, is a (weak) decoupling strategy where the linear and nonlinear parts of the reactive transport equation are first split and solved separately, and then combined to get the solution at a given time step.
